# Investigation of *Brucella melitensis* in Sable Antelope (*Hippotragus niger*) in South Africa

**DOI:** 10.3390/microorganisms8101494

**Published:** 2020-09-29

**Authors:** Barbara Glover, Malcolm Macfarlane, Roy Bengis, Jacques O’Dell, Johan Steyl, Henriette van Heerden, Darrell Abernethy

**Affiliations:** 1Department of Veterinary Tropical Diseases, Faculty of Veterinary Science, University of Pretoria, Onderstepoort, Pretoria 0110, South Africa; henriette.vanheerden@up.ac.za; 2Former Chief State Veterinarian of the Graaff Reinet State Vet Area, Eastern Cape 6280, South Africa; doc.mac@jabama.co.za; 3Former Chief State Veterinarian, Kruger National Park 1350, South Africa; roybengis009@gmail.com; 4Centre for Veterinary Wildlife Studies, Faculty of Veterinary Science, University of Pretoria, Onderstepoort, Pretoria 0110, South Africa; jacques.odell@up.ac.za (J.O.); johan.steyl@up.ac.za (J.S.); daa47@aber.ac.uk (D.A.); 5Department of Production Animal Studies, Faculty of Veterinary Science, University of Pretoria, Onderstepoort 0028, South Africa; 6Department of Paraclinical Sciences, Faculty of Veterinary Science, University of Pretoria, Onderstepoort 0028, South Africa; 7Aberystwyth School of Veterinary Science, Aberystwyth University, Aberystwyth, Wales SY23 3FL, UK

**Keywords:** brucellosis, *Brucella melitensis*, sable, wildlife, traceback, descriptive epidemiology, South Africa, disease outbreak, case studies

## Abstract

In this study, *Brucella melitensis* infection in sable antelope (*Hippotragus niger*) was investigated on two wildlife ranches in South Africa over a 12-year period in order to determine the origin of the outbreaks and the role of livestock in maintaining the disease. Retrospective data were obtained from farm records and interviews as well as samples tested from different disease scenarios and clinical settings. On one ranch, 10 of 74 sable tested seropositive for *B. melitensis* in 2004 but were certified clear of infection after no further brucellosis cases were detected following repeated serological tests and culling over a five-year period. Recrudescence occurred in 2013 (7 of 187 brucellosis positives) and in 2014 (one positive), with persistent, latent infection being the most reasonable explanation. In a second case study, linked to the first one through a common vendor, 15 of 80 sable tested positive in 2016, some five years after the acquisition of the animals from a putative source. *Brucella melitensis* biovar 1 and/or 3 were isolated from each outbreak on both ranches. Both outbreaks resulted in substantial losses for the owners, arising from testing and culling and significant resource expenditure by the state. The study identified the diagnostic challenges for identifying and resolving disease outbreaks in wildlife, the persistence of *B. melitensis* in sable, the risks associated with animal movements, and the need for a wildlife-sensitive disease control scheme. Although the actual source of infection could not be identified, the investigation points away from local livestock as a source of ongoing infection while the persistent infection is consistent with the disease circulating within small, ranched populations and being spread through the keeping and trading of high-value animals. The implications of the study findings to disease control in wildlife are discussed.

## 1. Introduction

Wildlife ranching, hunting, and tourism are major contributors to the green economies of southern Africa, although the practice and economic value of hunting remain controversial [1,2]. Sable antelope (*Hippotragus niger*) are a prized “high-value” species, favoured by both trophy hunters and ecotourists due their striking appearance and horns, and fetching high prices at auctions [3]. In the wild, they occur predominately in the wooded savannah regions of southern and east Africa, from southern Kenya to South Africa [4]. Their average lifespan is 19 years in the wild and 22 years in captivity [5]. They usually reach sexual maturity at five years [6], however, in some races, this might be as low as 24 months (Macfarlane, pers. comm.).

Brucellosis is a controlled disease in South Africa, focused primarily on mitigating the risk of bovine brucellosis (*Brucella abortus*) in cattle through a surveillance and disease control scheme and governed by relevant legislation (R.2483 of 9 Dec 1988 of the Animal Diseases Act 35 of 1984) [1]. *Brucella melitensis* is usually associated with small livestock and has been sporadically reported in sheep and goats (1965 [7], 1989 [8], and 1994 [9]). For both diseases, testing is voluntary unless infection has been detected or is suspected. Under the current scheme, “testing for bovine brucellosis is compulsory only for high-risk herds that have been confirmed as, or are suspected of being, infected. For all other herds and livestock owners, entering into a brucellosis testing scheme is voluntary” [10,11]. *Brucella melitensis*, however, is normally associated with small livestock, is not subject to routine surveillance, and testing only becomes compulsory for high-risk herds or during outbreaks [10]. A herd is labelled infected with brucellosis when found to be positive on serology tests or when *Brucella* spp. are confirmed on culture. On identification of positive samples, a whole herd is placed under quarantine followed by interventions to keep the farm free [11]; these activities are supported by the Animal Diseases Act, 1984 (Act No. 35 of 1984). The veterinary state actor (often the state veterinarian) investigates infected herds by identifying where the infection may have come from, identifying the extent of spread within and outside the herd, and by developing strategies for eradication. When infected is confirmed, the farmer is required by law to notify all the neighbours and all buyers who had purchased animals within the preceding 30 days [11]. According to the Animal Diseases Act, No. 35 of 1984, Section 11 [12], livestock farmers are required to take proactive measures to ensure their animals do not get infected. This includes not knowingly buying infected animals into a herd, reporting suspect infected animals to the responsible state veterinarian, and vaccinating livestock. Bovine heifers between four and eight months of age must be vaccinated once with either *B. abortus* biovar 1 strain 19 or RB51. In small ruminants, vaccination is undertaken with *B. melitensis* biovar 1 Rev 1 [13]. In wildlife, vaccination measures are not defined or recommended as yet, and farmers ultimately can only euthanise or cull infected animals. However, in disease control, the law has laid out proactive measures for control that can be applied to wildlife [11,14]. 

Brucellosis has been reported in southern African wildlife, particularly in sable antelope (*Hippotragus niger*), from 1972 [15], and from South Africa between 2005 and 2016 [6,16,17,18]. However, there are limited data on epidemiological investigations and origins of infections, particularly in breeding farms. 

Outbreak investigations are necessary to identify the source of brucellosis outbreaks, support control measures, and inform future actions [19]. This study assessed two different disease episodes, the first on a wildlife farm with *B. melitensis* infection identified in sable over a 10-year period (2004–2013) and the second with a single outbreak of multiple cases with clinical symptoms. Case study 2 was necessitated by a 2016 outbreak of *B. melitensis* biovar 1 infections in a group of sable that was characterised by hygromas, sometimes bilateral, over the carpal and tarsal joints. Both studies highlight the interventions taken, including disease surveillance measures, mitigation plans, and investigation of the putative source of infection and reinfection. 

## 2. Materials and Methods 

### 2.1. Study Sites

#### 2.1.1. Farm 1

Farm 1 was involved in extensive, mixed farming of sheep, goats, and wildlife. Prior to the first known outbreak, sable were maintained in three small herds of 10 to 12 cows and a bull (groups A, B, and C in Figure 1). There was also a variety of other ungulates (*n* = 4), including oryx (*Oryx gazelle)* and impala (*Aepyceros melampus*), and disease-free African buffalo (*Syncerus caffer*), as well as 700 sheep and goats. In 1999, the farmer acquired 31 roan antelope (*Hippotragus equinus*) from Farm 2 and seven sable from Farm 3 (Figure 1). The farmer exchanged 14 roan for 13 sable in July 2002 from Farm 2 and purchased nine sable from Farm 9 in October 2003 (see Figure 1), however, the only calf in the group died on arrival at the farm. The buffalo were certified negative for brucellosis in 1999 and sold in 2000.

#### 2.1.2. Farm 4

Farm 4 was equally involved in extensive, mixed farming, rearing sheep and goats, as well as wildlife. Prior to the outbreak, Farm 4 acquired nine sable (eight young cows and a 12-year bull) from Farm 2 in October 2011 and managed them in a bespoke management camp (Camp A, Figure 2) until 2013, when a new bull was purchased and the existing bull and heifers were moved to a new camp (Camp B). In early 2014, six of eight calves born the previous year died and malignant catarrhal fever was diagnosed. Deaths of calves continued in 2014, resulting in the farmer moving two cows to a camp (Camp C) alongside Camp B. All sable from Camps A and B were merged into a new group (Camp D) in October 2015 and the bull purchased from Farm 6 was moved to a separate camp (Camp E) with a single cow (see Figure 2).

### 2.2. Data Sources and Analysis

Data for this study were derived from farm records (Farm 1 and Farm 4), state veterinarian reports, on-farm interviews, and personal communication with the farmers. Farm, test sample numbers, and names of animal camps were anonymised in accordance with confidentiality agreements made with the farmers. For both case studies, data were analysed using Microsoft Excel 2016 64-bit edition [20]. 

### 2.3. Collection of Samples

All samples were collected according to the brucellosis scheme manual. The sable were immobilised by the local state veterinarian and blood samples collected from the jugular vein into serum tubes using 10 mL vacutainers with and without anticoagulant. Serum samples were sent directly to the nearest state veterinary laboratory for screening. Seropositive animals were euthanised, and lymph nodes, especially supramammary and retropharyngeal lymph nodes, were removed with the surrounding fat, packed in sterile disposable plastic bags, and transferred on ice to the state veterinary laboratories (Farm 1 and 4) and the Bacteriology BSL2+ laboratory at the Faculty of Veterinary Science, University of Pretoria (Farm 4). Fluid was collected from hygromas by extraction of 10 mL of the fluid through a syringe into vacutainers without anticoagulants. 

### 2.4. Testing of Samples

For Farm 1, serological samples were screened with the Rose Bengal test (RBT) and positives were confirmed with the complement fixation test (CFT). CFT titres of 49 IU or higher were defined as positive [21]. Tissue samples from slaughtered seropositive animals were sent to the provincial state veterinary laboratories for culture on Farrell’s medium (FM) and for histopathology. For Farm 4, sera were submitted in parallel to the Agriculture Research Council-Onderstepoort Veterinary Research (ARC-OVR) Laboratory, Onderstepoort, and to the Faculty of Veterinary Science, University of Pretoria (UP), South Africa. The farmer also submitted culled animals to the Pathology section at UP. Serological tests at UP consisted of screening by RBT and confirmation with IDEXX bovine brucellosis iELISA (IDEXX Laboratories, Inc., Westbrook, ME, USA), according to the manufacturer’s instructions with positive and negative controls. Positive reactors were defined as animals with agglutination for the RBT and iELISA with *S/P* value > 80. At necropsy, samples were taken from aborted foetuses, and from adults (lymph nodes, milk samples), which were then submitted to the UP Bacteriology BSL 2+ laboratory for culture on FM, as described by the World Organisation for Animal Health (OIE) [22], and modified CITA medium [21]. Suspect colonies were stained with modified Ziehl–Nielsen (Stamp) and oxidase, urease, and catalase tests were performed and confirmed to species level using PCR assays [23]. Biotyping (excluding phages) was undertaken as indicated by Edelsten [22] and OIE [23] at the ARC-OVR laboratory. 

### 2.5. Ethical Approval

Ethical approval was obtained from the Animal Ethics Committee of the University of Pretoria (V041-16) with Section 20 approval by the Directorate of Animal Health according to Act 35 of 1984, South Africa. 

## 3. Results

### 3.1. Farm 1

On 20 of February 2004, an abortion was reported by Farm 1 in a sable cow purchased from Farm 2. Blood samples, collected by the state veterinarian, were positive on RBT and CFT, and all sable were quarantined. The cow that aborted was re-tested, along with three other sable purchased at the same time from Farm 2. Two of the latter had lost calves, one of which presented with a uterine discharge and large bilateral hygromas on the stifle. Two were brucellosis-seropositive, including the primary animal and the animal with hygromas. All adult sable and some calves (*n* = 74) were tested by the end of June 2004, and nine were positive, all from Groups A and B (see Figure 1), including a sable bull and a calf. The latter died and *Brucella melitensis* biovar 3 was subsequently isolated from lymph node culture. Lymph node samples were also sent, by the farmer, to an unspecified laboratory overseas, and *B. melitensis* biovar 1 was identified. 

On Farm 1, all adult Angora goat ewes (*n* = 284), all merino ewes (*n* = 110) closest to the sable camps, three adult oryx, and one impala were tested for brucellosis by the end of June 2004. Livestock and wildlife on neighbouring farms were tested, including 400 sheep/goats on four farms, 23 kudu (*Tragelaphus strepsiceros*) on three farms, and 48 springbok (*Antidorcas marsupialis*) on three farms. All tested seronegative for brucellosis.

A mitigation plan was developed by the state veterinarian and the farmer and approved by the Directorate of Animal Health (Appendix A for information related to mitigation plan). It provided for repeated testing of the sable herd and release for sale of any animals with at least two negative brucellosis results. This led to a partial lifting of the quarantine on the farm in July 2004, only leaving the original quarantine of sable in place until further notice. 

In November 2004, the farmer was permitted to sell sable from Group C (see Figure 1) following two negative tests and isolation of the camp from May that year. Two females in Group C had calved normally and tested negative three weeks after calving. A group of 10 bulls and an undisclosed number of cows and calves were sold to Farm 18 later that month. The remaining animals in groups A and B at that time were all seronegative for brucellosis. After discussions with relevant authorities, it was decided to partially lift the quarantine notice on the sable in the farm with certain provisions in place. 

Four sable (two to four years old) had been sold in October 2003 to Farm 7 and were tested for brucellosis in November 2004 following the outbreak on Farm 1. A five-year-old cow with bilateral stifle hygromas and deformed hooves tested positive. The cow was originally bought from Farm 2, had aborted at least once, and was lame. *B. melitensis* biovar 3 was isolated from hygroma fluid and lymph nodes. The farm was placed under quarantine in November 2004, which persisted until November 2005, during which all animals on the farm tested negative for brucellosis.

By December 2007, all infected sable on Farm 1 had been destroyed and serological testing of the remaining sable had been undertaken, with negative results. Ongoing monitoring was conducted, including testing of animals at approximately nine months of age when they were immobilised to be microchipped, or upon translocation. The quarantine was lifted in December 2007.

By early 2013, the animals on Farm 1 had increased to 187 sable, approximately 4000 sheep, and 25 buffaloes. The sable were divided among six camps; in February, a sable bull in Camp 3 developed acute hindquarter paresis, and swollen carpal joints were observed at examination. It was out of a sable cow of unknown identification, but which had been present in the first outbreak. The bull was seropositive to the RBT and CFT and was subsequently destroyed; *B. melitensis* biovar 1 was isolated from tissue culture. Over the following six weeks, over 66 sable were tested across six camps (see Figure 1), including 29 cows that were present during the first outbreak. Six animals tested positive, including a lame cow and her offspring, which was a pregnant heifer at that time. *B. melitensis* biovar 1 was isolated from five of the seropositive animals. Trace testing revealed three positive sable cows on a farm (Farm 8) that had purchased them in late 2012 from Farm 1 (see Figure 1).

A mitigation plan was also developed for this outbreak, which enabled Farmer 1 to proceed with his wildlife farming business whilst resolving the brucellosis incident. In camps in which seropositive sable were disclosed, animals were only classified as negative following repeated seronegative test results at three-month intervals, including all calves up to one year old. Over 193 animal tests were conducted and, by June 2013, all sable present on the farm and those forward-traced had been tested, as well as 358 other animals, all of which tested negative.

In April 2014, a non-pregnant heifer in Camp 5 tested positive on RBT and CFT; it was destroyed, and *B. melitensis* biovar 1 was isolated. This was the first positive animal in Camp 5 and further serological testing proved negative. Testing was conducted on neighbouring farms (*n* = 290 sheep and goats), all with negative results. Monitoring of the herd continued through 2015 and 2016, with no further reactors identified. In accordance with the mitigation plan, these animals could then be sold with a pre-movement test while purchased bulls would be tested after arrival and all heifers retested post-calving. 

Brucellosis was also detected in 2013 on farms associated with Farm 1 and following sale of sable that had been present during the 2004 outbreak but had repeatedly tested negative. A sable cow had been sold in 2012 to Farm 8, where it tested positive following non-specific lameness, swollen hocks, and abortion. It was positive along with three contact animals, and *B. melitensis* biovar 1 was again isolated. A sable cow, sold to Farm 12 in 2007 and tested in February 2013 following non-specific lameness, was also positive on serology and culture. 

In summary, the farm prevalence for Outbreak 1 on this ranch was 13.2%, with infected animals detected from four camps. Biotyping results were obtained for eight animals, with the dominant strain being *B. melitensis* biovar 1, although one positive sable was identified as *B. melitensis* biovar 3. Seventy percent (7/10) of the positive animals were either purchased from Farm 2 or were offspring of animals bought from Farm 2. In Outbreak 2, 11 (27%) of the 41 animals tested were positive. Of the positives, nine (82%) were female whilst two (28%) were calves. Biotyping results showed the re-circulation of the *B. melitensis* biovar 1 strain in the populations. Eight (72.7%) positives were present in Outbreak 1, two (18.2%) were born on the farm post-Outbreak 1, and one (9.1%) was bought in from Farm 2 just before the second outbreak. In the third outbreak, one sable on the farm from a total of 388 sable tested was positive. Additionally, five sable sold to Farm 12 also tested negative, bringing it to a total of 6 (1.3%). The positive animals were all cows and were present in Outbreak 2. Biotyping results confirmed the presence of *B. melitensis* biovar 1.

### 3.2. Farm 4

In February 2016, brucellosis was diagnosed on serology in Camp D following late calving. Of 63 sable on the farm, four additional animals were seropositive: two cows from Camp D, which exhibited hygromas; the bull in Camp E; and one of two cows in Camp C. *Brucella melitensis* biovar 1 was isolated from one of the cows. All positive animals were destroyed, as well as the (seronegative) cow in Camp C. A total of 59 buffalo were tested and all proved negative. The farm was subsequently quarantined and serological testing of sheep and goats on neighbouring farms implemented, all of which were negative. All surviving sable were retested in June 2016, at which time a further 15 were seropositive. In 2018, the farmer destroyed all sable on the farm (see Figure 1). Following the diagnosis of brucellosis on Farm 4, the farmer notified farms from which he had acquired sable (Farms 2 and 6) and those to whom he had sold animals (Farm 5). All tests on Farms 2 and 6 were negative, while two cows on Farm 5 tested positive and were destroyed, along with their offspring.

In summary, the overall farm prevalence was 17.4% (19 of 109 positive), with nine positives being female whilst six were calves. Most (68.4%) of the positives were identified in the camp of animals purchased from Farm 2, while four had been sold to Farm 5 prior to the outbreak. 

## 4. Discussion

*Brucella abortus* is well recognised in wildlife, among multiple continents, where it has a similar pathobiology to that of livestock and can negatively impact population fecundity [24,25,26,27]. By contrast, *B. melitensis* has only been sporadically reported, notably in various European ungulates [28], and little is known about the epidemiology or transmission dynamics of infection. Its presence in wildlife may be underestimated due to the inability of serological surveys to distinguish between *B. abortus*, *B. melitensis*, and *Brucella suis* [28]. *B. melitensis* has been sporadically reported in sable in southern and South Africa from 1972 [9,10,20,21], however, its occurrence has been minimally studied, and origins of infections, particularly in breeding farms, remain a mystery.

The wildlife resources of South Africa play an important but disputed role in its green economy, contributing to employment, biodiversity, and conservation. They are divided between state-owned game reserves and the wildlife industry, which includes wildlife ranching (mainly breeding and live animal sales), wildlife activities (mainly wildlife viewing and hunting), and wildlife products (mainly venison and hides) [29]. In 2014, the number of private ranches was estimated to exceed 9000, occupying a total area of 170,419 km^2^ [30]. The sector has experienced significant growth, with rates of 5.6 to 6.75% recorded per annum since conditional private ownership of wildlife was granted in 1991. However, despite such growth, the sector’s contribution to transformation, empowerment, and conservation remain controversial [29]. Core to the industry is the breeding and trading of animals, with one 2010 estimate indicating that over 167,000 animals were translocated that year [31]. Given this degree of activity, the potential for disease transmission associated with the movement of animals remains high.

Brucellosis infections in wildlife are often linked with livestock–wildlife contact [32,33,34,35] and disease control is challenging, requiring regular monitoring and sampling, which is hampered by the environment in which wildlife are maintained and expense involved in the use of helicopters and chemical immobilisation [36]. Adopting the approaches used to address disease control in livestock is often impracticable and requires unique, targeted approaches in controlling brucellosis in wildlife, especially on farms with both livestock and wildlife.

This study was the first to undertake an epidemiological investigation of *B. melitensis* using retrospective data and current outbreaks. Key objectives included confirming the organism involved and then identifying the origin of infection and factors that facilitated its spread. Infection was confirmed through culture and biotyping, with the causative organism identified as *B. melitensis* biovar 1 and 3 in Outbreak 1 and 2, while *B. melitensis* biovar 1 was identified in Outbreak 3 on Farm 1. Both have been previously been reported in South African livestock; *B. melitensis* biovar 2 and 3 were isolated from cattle [37] and B. melitensis biovar 1 from sheep [8].

The investigation revealed multiple factors that impact upon brucellosis control among ranched sable, with implications for other wildlife species as well. First, it revealed the ability of sable to maintain infection within the population for at least 12 years, without apparent spillover from local livestock. Outbreaks outside of this investigation, in 2015 and 2016, were attributed to spillover from livestock to sable [38], but no evidence (concomitant infection in livestock) was demonstrated at that time to support such a conclusion. Such occurrences have been reported elsewhere, for example, in ibex (*Capra ibex*) and chamois (*Rupicapra rupicapra*) [39], and in wild goats (*Capra pyrenaica*) [40]. In the outbreaks investigated in this study, extensive and repeated serosurveillance of small ruminants on the infected ranches or on neighbouring farms failed to detect any indication of infection. This suggests spillover was improbable, although it could not be entirely excluded. This was further supported by whole genome sequencing, which revealed that the *B. melitensis* isolates from sable antelope on Farms 1, 2, and 4 clustered in a unique clade separate from that isolated from small ruminants in South Africa (B. Glover, unpublished data). 

Second, the nature of sable ranching disclosed during the inquiry was highly conducive to disease spread, with multiple between-ranch movements, the creation of multiple camps within ranches, and the associated between-camp movements. Sable were usually confined to camps for breeding or management purposes, but this had no protective effect on infection, as revealed by the distribution of seropositive animals across multiple camps. Farm 2 was the source of animals for both Farms 1 and 4 and, notably, at the time of the outbreak on Farm 1, was under quarantine for brucellosis detected in African buffalo on the property. All the positive animals on Farm 1, with the exception of one bull acquired from Farm 6, were sourced from Farm 2. However, the sable tested on Farm 2 were negative, thus suggesting testing was incomplete or that the infected animals had already left the ranch. 

The original source of infection was not identified by the investigation, but it is possible that infection entered South Africa with the importation of sable from other African countries, most notably Zambia. Sable from western Zambia are highly prized in South Africa as they are heavier bodied, with larger horns. Accordingly, ranchers cross-breed Matsetsi sable, a race indigenous to South Africa, with the western Zambian sable in order to produce offspring more phenotypically resembling the nearly extinct giant sable [4]. These attempts ultimately failed as it was later established through evolutionary genomics that the giant sable and the two crossed sables were different species [4]. Importations did occur but were controversial and disputed, and illegal importations were reported in the media [41,42], eventually being addressed through court action in 2014. As a result of this, the veterinary authorities in South Africa commissioned an independent risk analysis [43], which, although identifying the risk of introducing *B. melitensis* through importation as low to very low, raised substantial questions. These included the lack of disease surveillance and control measures in the exporting country, the paucity of knowledge of the disease in sable, and the lack of validated tests [44]. Significantly, the report identified infection in African buffalo that persisted through pre-export procedures and noted that one sable bull in a lot of 150 due to be exported to South Africa had tested negative to the CFT one month after testing positive to both RBT and competitive Enzyme-Linked Immunosorbent Assay (ELISA) [44]. It is also notable that there was a striking temporal correlation between the occurrence of *B. melitensis* infections in wildlife in Zambia on the OIE database [45] and the periods of smuggling arrests and subsequent periods of the introduction of *B. melitensis* into South African sable industry [46]. 

One cannot, therefore, exclude the possibility that *B. melitensis* was introduced with an earlier importation; the findings on Farm 1 demonstrate that infection can persist many years, despite systematic testing and culling. Similar to infection in cattle [47], *B. melitensis* persists in sable populations through heifers that test positive for brucellosis only when they are sexually mature. In Outbreak 1, the rancher avoided testing calves due to the risk from chemical immobilisation. Some of these calves then tested positive four to five years later, which is the age at which some sables become sexually mature [6]. Latency is described by Department of Agriculture, Land Reform and Rural Development (DALRRD) as the “two-year breakdown” syndrome, where infection cleared from a farm reappears within two years in 2 to 20% of reported outbreaks [11]. 

A third conclusion of the investigation was the possibility of sexual transmission to sable cows and then to subsequent offspring. Infections were first identified in sable bulls in both Outbreak 2 and 3; they were the index cases, the putative source of infection brought for the ranches, with *B. melitensis* also being isolated from the testes of bulls in Outbreak 3. This is consistent with the potential venereal transmission, as has been reported in the Alpine ibex [48]. The Bovine Brucellosis Manual of South Africa stipulates that “should semen from an infected sable bull be used for artificial insemination, the risk of spread of the disease is great, but if used for natural service sable bulls do not appear to play a role in the spread of the disease” [11]. However, a recent study of risk factors in brucellosis transmission in cattle reported farms that use sable bulls from farms with positive/unknown *Brucella* status had a higher risk of infection than those that use artificial insemination [49]. One bull was used for breeding purposes in Farm 2 before being sold to Farm 4 for a similar role. Although never tested, it presented with the typical signs of arthritis and lameness before death and was in the camp with multiple positive infections. 

A fourth finding was the paucity of data on sable purchases and movements in farm records that adversely affected the investigation and the ability to fully pinpoint the actual sources of infection and reinfection in Farm 1 and Farm 4. The absence of a wildlife brucellosis scheme presents significant challenges to controlling the disease and is essential if brucellosis is to be controlled, not only in sable but in other wildlife species. Arguably, the bovine brucellosis scheme can be applied to wildlife, but the dynamics are different and impractical in wildlife. The current bovine scheme requires repeated testing of suspect herds, which works readily for livestock but is both impractical and prohibitively expensive when applied to free-ranging wildlife. Aside from the inherent dangers that chemical immobilisation poses to wildlife, helicopter costs, tranquilisers, and veterinary costs make repeated testing unfeasible. Any wildlife scheme should incorporate good records (movements, interventions, census data) and regular, opportunistic monitoring (sampling at translocation, sales, hunting) to build up a history of disease evaluation and control on each ranch, which can be supplemented by purposive sampling and pre-movement testing when required. Cognisance must be taken of the huge costs of disease control, which are currently borne by the owner, with no compensation for destroyed animals or contribution towards sampling costs. This supports the veterinary authority’s policy of having the local veterinarian develop a strategy for control along with the rancher and local state veterinarian. Such collaboration is to be commended and should be further enhanced. 

Serological tests must be validated as these underpin any surveillance or control system and scheme guidelines should utilise probability sampling rather than whole-herd evaluations to determine a path to certified freedom of disease. Finally, standardised investigation reports should be completed and, as indicated in these outbreaks, should be retained for long-term benefits.

## 5. Conclusions

The role of sable as potential reservoirs of *B melitensis* for human and domestic livestock appears to be strongly supported by this study but remains to be properly defined, given the lack of epidemiological data. The huge costs of control once disease is suspected are all borne by the owner, necessitating the need for a scheme that protects both wildlife and industry and meets the responsibility of governments to protect human and animal health. 

Concerns remain that there are possibly other erosive or trade-sensitive diseases in wildlife, but apart from buffalo, there are no official animal disease control or monitoring strategies for wildlife in South Africa. This study contributes additional data to what is currently known about sable brucellosis and identifies the need for further controlled studies on wildlife brucellosis as a basis for informed control.

## Figures and Tables

**Figure 1 microorganisms-08-01494-f001:**
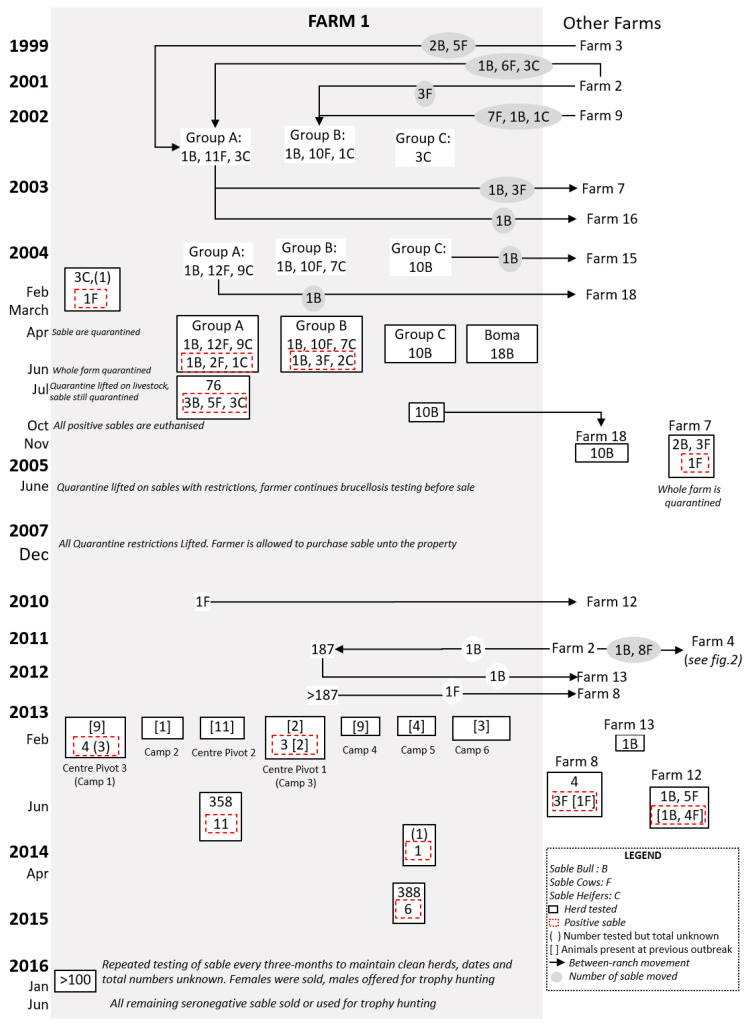
Flow diagram of postulated brucellosis spread in Farm 1 from 1999 to 2016. Seropositive and seronegative herds are indicated at the time of testing. The sable (bull (B), cow (F), or heifers (C) on Farm 1 with movements (indicated by the grey circle) and groups are shown from 1999 to 2003 before the first brucellosis-seropositive animals were tested in 2004 (indicated by stipulated red box).

**Figure 2 microorganisms-08-01494-f002:**
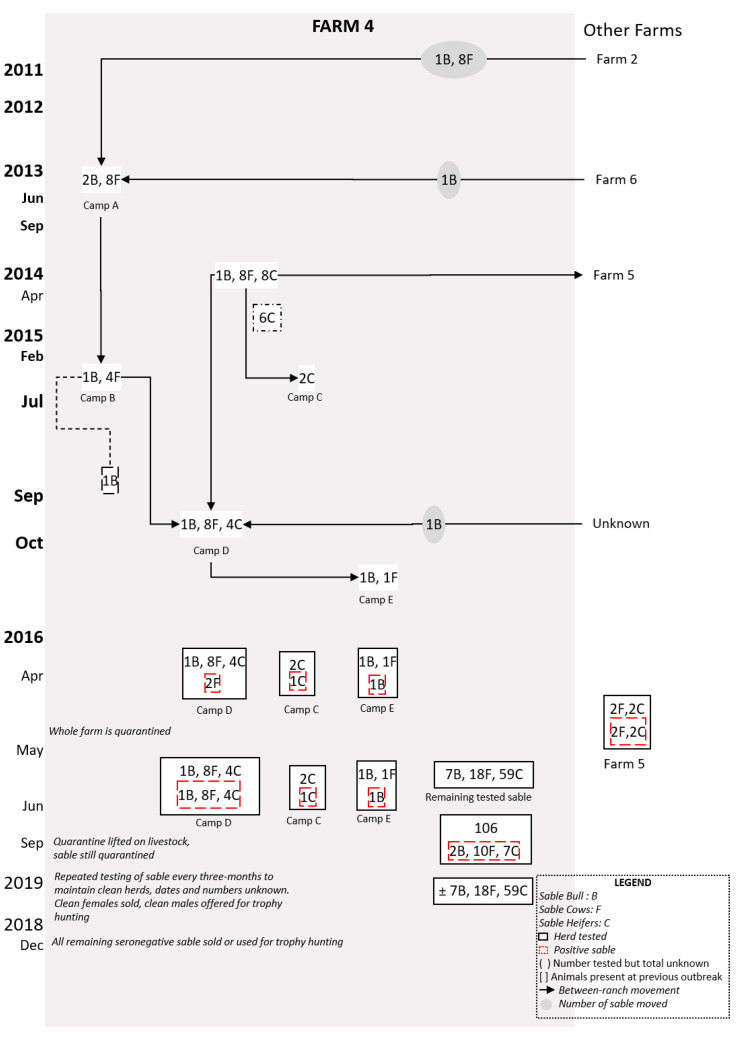
Flow diagram of postulated brucellosis spread in Farm 4 from 2011 to 2018. The sable (bull (B), cow (F), or heifers (C) on Farm 4 with movements (indicated by the grey circle) and groups are shown from 2011 to 2015 before the first brucellosis-seropositive animals were tested in 2016 (indicated by stipulated red box).

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
