# Peer review of "Investigation of *Brucella melitensis* in Sable Antelope (*Hippotragus niger*) in South Africa"

_microorganisms, 2020, doi:10.3390/microorganisms8101494_

Round 1
Reviewer 1 Report
The manuscript by Glover et al. describes two episodes of brucellosis outbreak in sable that occurred in two farms in South Africa. To date, there are limited data describing the role of wildlife in the spread of brucellosis in Africa and the diversity of Brucella spp., particularly in the central and southern parts of the continent. This study demonstrate that farmed sables can contribute to the dissemination of the disease within farms.
The authors attempted to describe the route of dissemination of B. melitensis in two establishments over several year period. The study provides some interesting insights into the dynamics of the disease in South Africa, but in the current version the manuscript it is incomplete and requires multiple modifications.
Major comments:
- Figure 1 and 2 - the figures are overcomplicated (in particular Fig 1) and must be simplified. The diagrams are not self-explanatory and require time to understand the connections between farm feeders, receivers and the animals transferred and/tested. Moreover, the numbers are inconsistent and it is impossible to understand if these inconsistencies come from the number of animals born or from the trade with unknown source. Moreover the '?' remain unexplained and in some cases there are numbers in brackets - again with no explanation. The figures must be re-designed to unable the reader to understand the flow of the diagram without the need to read the entire manuscript.
- The study is generally poor in data and could be summarised in one paragraph. Additional information should be supplied. I suggest adding a table that describes in more detail the positive cases isolated in this study. The fields could contain information such as origin of the animal, symptoms, positivity to specific tests, organism isolated etc. Moreover, in the discussion the authors state that the strains were sequenced. Why is this data not included? In modern epidemiology, the sequencing data provide essential means to track and trace the origins of infection and should be incorporated to provide more strength to the conclusions reached by the authors.
- In introduction, the authors do not state the aims of the study clearly. The aims are however mentioned in the discussion L 261- L262. These include: identifying the origin of infection (i), and factors facilitation spread (ii). i) Unfortunately, while there is a correlation between the animal seller (farm 2) and subsequent positivity of the purchased animals, there is no evidence that this farm was the source of the infection and therefore the conclusion that points at this farm cannot be made. It seems unusual that from the farm where all animals tested negative, animals infected with two different B. melitensis biovars (3 and 1) would have been purchased. The authors should discuss it in more details. ii) The factors facilitating the spread of the disease are not clearly described.
- L86-88 - in material and methods please describe in details which mitigations plans and surveillance methods were introduced by the farmers.
Reviewer 2 Report
The MS by Barbara Glover et al. 2020 describves an investigation of Brucella melitensis infection in sable antelope in South Africa.
This study is important to evaluate the role of sable as potential reservoirs of the bacteria for human and domestic livestock. Overall, the MS is perfect. The information is well written and the data is well presented. So I recommend this MS to be accepted in Microorganisms.
Author Response
Thank you for the review!
Round 2
Reviewer 1 Report
The authors modified the manuscript according to the suggestions or provided a satisfactory explanation of the issues that had been raised.
The manuscript still contains minor mistakes and should be proof-read and corrected before publishing.
e.g.
L154 "indicated by [22] and OIE [23] at the ARC-OVR laboratory" - provide the actual title before the reference 22.
L185 insert a space in Bat - B at
L 227-238 - insert spaces between text and brackets
Author Response
The authors thank the reviewer for the second round of review. The document has been proof-read for errors and these have been corrected.
